# dEBORA: Efficient Bilevel Optimization-based low-Rank Adaptation

**Emanuele Zangrando[1], Sara Venturini[2], Francesco Rinaldi[3], Francesco Tudisco[1,4,5]**

[1]Gran Sasso Science Institute, L'Aquila, Italy
[2]MOBS Lab, Northeastern University, Boston, US
[3]Department of Mathematics, University of Padova, Padova, Italy
[4]School of Mathematics and Maxwell Institute, University of Edinburgh, UK
[5]Miniml.AI Ltd, UK

## ABSTRACT

Low-rank adaptation methods are a popular approach for parameter-efficient fine-tuning of large-scale neural networks. However, selecting the optimal rank for each layer remains a challenging problem that significantly affects both performance and efficiency. In this paper, we introduce a novel bilevel optimization strategy that simultaneously trains both matrix and tensor low-rank adapters, dynamically selecting the optimal rank for each layer. Our method avoids the use of implicit differentiation in the computation of the hypergradient, and integrates a stochastic away-step variant of the Frank-Wolfe algorithm, eliminating the need for projection and providing identifiability guarantees of the optimal rank structure. This results in a highly efficient and cost-effective training scheme that adaptively allocates the parameter budget across the network layers. On top of a detailed theoretical analysis of the method, we provide different numerical experiments showcasing its effectiveness.

## 1 INTRODUCTION

Parameter-efficient fine-tuning methods have become essential for adapting large-scale pre-trained models to various downstream tasks without incurring prohibitive computational costs. Low-rank adaptation (LoRA) methods, which involve the addition of low-rank matrices to the existing weights of neural networks, have emerged as a popular solution due to their efficiency and simplicity. These methods keep the main model weights frozen, significantly reducing the number of trainable parameters and making the fine-tuning process more resource-efficient.

However, a persistent challenge in this area is the selection of the optimal rank for each low-rank adapter, which is crucial for balancing performance and efficiency. Traditional LoRA methods often rely on a static rank configuration across all layers, leading to suboptimal results. Recent approaches such as DyLoRA (Valipour et al., 2023) and AdaLoRA (Zhang et al., 2023) have attempted to address this by dynamically adapting the rank during training, but they still require considerable heuristic tuning or involve expensive steps that can limit their applicability and generalizability.

In this paper, we propose dEBORA (Efficient Bilevel Optimization-based low-Rank Adaptation), a novel approach that leverages bilevel optimization to simultaneously train matrix and tensor low-rank adapters while dynamically selecting the optimal rank for each layer. Unlike previous bilevel optimization-based methods for low-rank adaptation (Qiang et al., 2024), dEBORA eliminates the need for implicit differentiation in the computation of the hypergradient, thereby hugely simplifying the resources required by the optimization process. The upper-level rank-adaptation step is performed by means of a stochastic away-step variant of the Frank-Wolfe algorithm, which not only removes the need for costly projection steps but also provides identifiability guarantees of the optimal rank structure.

We provide a detailed theoretical analysis of our method, establishing its convergence properties and optimality guarantees. Additionally, we conduct extensive numerical experiments across a range of benchmarks, including natural language understanding and generation tasks, to demonstrate the

effectiveness of dEBORA. Our results show that dEBORA outperforms existing low-rank adaptation methods in both efficiency and performance, particularly in settings with stringent parameter budgets.

Our method addresses the key limitations of existing low-rank adaptation techniques in several ways. First, by dynamically adjusting the rank for each layer during training, dEBORA ensures an adaptive allocation of the parameter budget across the network, thereby improving both computational efficiency and model performance. Second, the use of bilevel optimization provides a principled framework for handling the interaction between the model parameters and the adaptation budget hyperparameters, resulting in a more robust and theoretically grounded approach to low-rank adaptation.

The rest of this paper is organized as follows: Section 2 reviews related work on parameter-efficient fine-tuning and low-rank adaptation methods. Section 3 presents the proposed bilevel optimization framework and the dynamic rank adaptation mechanism, while Section 4 the specific structure of the problem is used to create an efficient closed-form approximation for the hypergradient. Section 5 and Section 6, introduce the stochastic version of the away-step Frank-Wolfe algorithm for solving the upper-level problem, along with its theoretical properties. Section 7 details the experimental setup and results. Finally, Section 8 concludes the paper and discusses potential future directions.

## 2 RELATED WORK

**Parameter-Efficient Fine-Tuning.** Large-scale pre-trained models have demonstrated significant improvements across numerous tasks. However, fine-tuning these models for specific downstream tasks often involves updating millions, if not billions, of parameters, leading to substantial computational and memory overhead. This has motivated the development of parameter-efficient fine-tuning (PEFT) techniques, which aim to reduce the number of trainable parameters while maintaining or even enhancing the performance of the model.

One of the seminal works in this area is Low-Rank Adaptation (LoRA) (Hu et al., 2021), which introduces low-rank incremental matrices to the frozen pre-trained weights, significantly reducing the trainable parameters. Inspired by recent findings in e.g. (Li et al., 2018) and (Aghajanyan et al., 2021), which suggest that pre-trained models possess a low intrinsic dimension, LoRA leverages this property to achieve efficient adaptation without compromising performance.

Several variants of LoRA have been proposed to further improve its efficiency and adaptability. DyLoRA (Valipour et al., 2023) dynamically adjusts the ranks of the low-rank matrices during training based on the importance of learned representations, while QLoRA (Dettmers et al., 2024) introduces quantization techniques to reduce the memory footprint of LoRA, making fine-tuning accessible on hardware with even more limited resources. Another extension is LoraHub (Huang et al., 2023), which facilitates the modular composition of multiple LoRA modules across different tasks, enhancing the cross-task generalization capability of fine-tuned models.

Similar to our approach, AdaLoRA (Zhang et al., 2023) takes this a step further by adaptively allocating the parameter budget based on the importance of different modules within the model. By parameterizing the low-rank updates using a singular value decomposition (SVD) framework, AdaLoRA iteratively prunes less significant singular values, thus optimizing the allocation of the limited parameter budget across the network.

BiLoRA (Qiang et al., 2024) employs a bi-level optimization framework, where the learning of pseudo singular values and vectors is decoupled and assigned to different levels of the optimization hierarchy. This separation mitigates the risk of overfitting by allowing each component to be optimized independently on separate subsets of the training data. This approach is inspired by differentiable architecture search (DARTS) (Liu et al., 2019), where the architecture and weights are optimized on different datasets, preventing overfitting to any particular training set. Unlike the proposed dEBORA, BiLoRA computes the hypergradients "directly" using implicit differentiation, resulting in high computational demand.

**Low-Rank Methods for Pre-Training.** Low-rank methods have also been successfully applied during the pre-training and training phases of neural networks, capitalizing on the observation that large models often possess a low intrinsic dimensionality. Techniques like Pufferfish (Wang et al., 2021), intrinsic dimension reduction (Aghajanyan et al., 2020), and DLRT (Schotthöfer et al., 2022) reduce the number of parameters during training, potentially improving both model efficiency and

generalization. Recent developments such as (Zangrando et al., 2024) use Riemmanian optimization to explore the parameter space, dynamically adapting the rank during training and ensuring model accuracy and substantial memory reduction, even with initially incorrect rank estimates. ReLoRA (Lialin et al., 2023) introduces a method for training large models efficiently by applying multiple low-rank updates, achieving significant memory savings while maintaining performance. GaLore (Zhao et al., 2024) is based on projecting the gradients onto a low-rank subspace, allowing for memory reductions without sacrificing model accuracy.

**Bilevel Optimization in Deep Learning.** Bilevel optimization, with its origins in classical optimization theory (Colson et al., 2007), has gained traction in machine learning for a variety of applications, including hyperparameter optimization (Franceschi et al., 2018), meta-learning (Finn et al., 2017), and neural architecture search (Liu et al., 2019). The core challenge in bilevel optimization lies in the computation of the hypergradient, which captures the dependence of the upper-level objective on the lower-level variables. This often scales the complexity of first-order optimization methods to that of second-order ones, making it computationally prohibitive for large-scale problems.

To address this challenge, numerous techniques have been proposed to approximate the hypergradient efficiently. Backpropagation through time (BPTT) (Franceschi et al., 2017) is one such approach, along with methods that use efficient inverse Hessian approximations (Lorraine et al., 2020) and approximate implicit differentiation (AID) techniques (Grazzi et al., 2020). These methods have made bilevel optimization feasible for high-dimensional problems, albeit with trade-offs in terms of computational accuracy and convergence speed.

Our work introduces a novel bilevel optimization strategy for parameter-efficient fine-tuning, specifically tailored for low-rank adaptation. By avoiding implicit differentiation and integrating a stochastic away-step variant of the Frank-Wolfe algorithm, our approach eliminates the need for costly gradient propagations and offers a more scalable solution. This framework allows for adaptive rank selection across network layers, providing a robust and efficient mechanism for fine-tuning.

## 3    LOW-RANK ADAPTATION VIA BILEVEL OPTIMIZATION

In this section, we introduce the bilevel optimization framework for fine-tuning low-rank adapters in tensor format, which provides a parameter-efficient adaptation strategy for large pre-trained models.

Consider the adaptation of pre-trained model weights $\mathcal{W}_0 \in \mathbb{R}^{n \times n}$ through a low-rank update $\Psi = USV^\top$, where $U, V \in \mathbb{R}^{n \times r}$ are the low-rank basis matrices, $S$ is a diagonal matrix containing the singular values $s_i$, and $r$ is the rank of the adaptation. This low-rank formulation can be equivalently expressed as

$$\Psi = USV^\top = \sum_{i=1}^{r} s_i\, u_i \otimes v_i,$$

where $u_i$ and $v_i$ denote the $i$-th columns of $U$ and $V$, respectively, and $\otimes$ refers to the outer product of tensors. The rank $r$ directly controls the number of trainable parameters, making this formulation particularly suitable for rank-adaptive parameter-efficient fine-tuning (Zhang et al., 2023).

This low-rank adaptation naturally extends to tensor-based models using the CP (canonical polyadic) decomposition. For a layer weight tensor with $d$ modes, $\mathcal{W}_0 \in \mathbb{R}^{n_1 \times \cdots \times n_d}$, the low-rank adaptation is defined as:

$$\mathcal{W}_0 + \Psi(\mathcal{B}, s),$$
$$\Psi(\mathcal{B}, s) = \sum_{i=1}^{r} s_i\, u_i^{(1)} \otimes \cdots \otimes u_i^{(d)},$$

where $\mathcal{B} = (U^{(1)}, \ldots, U^{(d)})$ represents the collection of low-rank matrices $U^{(i)} \in \mathbb{R}^{n_i \times r}$, $u_i^{(j)}$ denotes the $i$-th column of the $j$-th basis matrix, and $s = (s_1, \ldots, s_r) \geq 0$ is a vector of nonnegative

parameters representing a form of tensor singular values. In this setting, the low-rank adaptation is compactly represented by the basis matrices $\mathcal{B}$ and the vector of scaling factors $s$.

To fine-tune $\Psi(\mathcal{B}, s)$ while simultaneously minimizing the rank $r$, we introduce a bilevel optimization strategy. Given a fine-tuning loss function $\ell(\mathcal{B}, s; x, y)$ and a dataset split into two parts $\mathcal{D}_1$ and $\mathcal{D}_2$, we consider objective functions for the two data sets

$$f_i(\mathcal{B}, s) = \frac{1}{|\mathcal{D}_i|} \sum_{(x,y) \in \mathcal{D}_i} \ell(\mathcal{B}, s; x, y), \qquad i = 1, 2.$$

The fine-tuning process is then formulated as the following bilevel optimization problem:

$$
\begin{aligned}
\text{(upper-level)} \quad & \min_{s \in \mathbb{R}^r : s \geq 0, \|s\|_1 \leq \tau} f_1(\mathcal{B}^*(s), s) \\
\text{(lower-level)} \quad & \text{s.t.} \quad \mathcal{B}^*(s) \in \underset{\mathcal{B} \in \mathcal{V} \subseteq \mathbb{R}^{n_1 \times r} \times \cdots \times \mathbb{R}^{n_d \times r}}{\arg\min} f_2(\mathcal{B}, s),
\end{aligned}
\tag{1}
$$

where $\tau > 0$ is a regularization parameter that controls the sparsity of the singular values. The bilevel nature of this problem arises from the fact that the optimal basis matrices $\mathcal{B}^*(s)$ for a given $s$ are determined by minimizing a second objective function, $f_2(\mathcal{B}, s)$. Moreover, given the nature of the parameterization $\Psi$, none of the objective functions is in general convex, even if $\ell$ is.

A critical aspect of solving bilevel optimization problems is computing the hypergradient, which captures how changes in the upper-level parameters $s$ affect the lower-level solution $\mathcal{B}^*(s)$. By the chain rule, the hypergradient of the upper-level objective $f_1(\mathcal{B}^*(s), s)$ is given by:

$$\frac{d}{ds} f_1(\mathcal{B}^*(s), s) = \partial_{\mathcal{B}} f_1(\mathcal{B}^*(s), s) \, \partial_s \mathcal{B}^*(s) + \partial_s f_1(\mathcal{B}^*(s), s).\tag{2}$$

The key computational challenge lies in determining $\partial_s \mathcal{B}^*(s)$, which represents how the optimal basis $\mathcal{B}^*(s)$ changes with $s$. Using first-order stationarity conditions, this dependency is governed by the implicit gradient system:

$$\partial_{\mathcal{B}}^2 f_2(\mathcal{B}^*(s), s) \, \partial_s \mathcal{B}^*(s) = -\partial_s \partial_{\mathcal{B}} f_2(\mathcal{B}^*(s), s).\tag{3}$$

Solving this linear system is necessary to compute the implicit gradient in Equation (2).

Given the computational complexity of solving the implicit gradient system exactly, we explore efficient approximation techniques that exploit the structure of the problem. As discussed in Section 2, various methods, such as inverse Hessian approximation and conjugate gradient techniques, have been proposed in the literature. In the following section, we introduce a new closed-form approximation that leverages the tensor structure of the problem to significantly reduce computational overhead while maintaining accuracy.

For the sake of simplicity, we decided to limit the theoretical discussion to the Euclidean case. However, everything transfers with minor adjustments to the Riemannian setting by interpreting all the derivatives through their Riemannian counterpart. A similar derivation is done for example in (Li & Ma, 2024) and the computations in our case would be similar to those presented there. In particular, if we have Riemannian manifold only in the lower-level problem (the case we considered in our experiments), the stationarity condition should be interpreted as $\partial_{\mathcal{B}} f_2|_{\mathcal{B}^*(s), s} = 0$ where $\partial_{\mathcal{B}} f_2|_{\mathcal{B}^*(s), s}$ is now the differential restricted to the tangent space $T_{\mathcal{B}^*(s)} \mathcal{V}$ to the manifold $\mathcal{V}$ at the point $\mathcal{B}^*(s)$ (with $\mathcal{V}$ either the Steifel or the oblique manifold in our case). The implicit gradient equation (3) is derived in the same way by interpreting the Hessian as the Riemannian Hessian on $\mathcal{V}$.

## 4 Efficient Hypergradient Approximation

As discussed in the previous section, the primary challenge in solving the bilevel optimization problem in Equation (1) lies in the fact that first-order methods require second-order information, particularly the calculation of the implicit derivative, which is obtained as the solution to Equation (3). While this can be computationally intensive, structured numerical approximations of the Hessian $\partial_{\mathcal{B}}^2 f_2^*$ (such as diagonal or low-rank approximations) have been employed in prior work (Zhang et al., 2022; Lorraine et al., 2020) to simplify the hypergradient computation. Nonetheless, these approaches come with computational overhead.

In our setting, we exploit the specific parameterization structure of the problem to derive an efficient closed-form approximation for the hypergradient. This leads to a computationally tractable solution without fully solving the implicit gradient system. The following theorem formalizes this result. For brevity, we will use a superscript $\cdot^*$ to denote the evaluation of variables and functions on the optimal pair $(\mathcal{B}^*(s), s)$, solution to the lower-level problem.

**Theorem 4.1** (Hypergradient Approximation). *Denote by $L_2(\Psi(\mathcal{B}, s)) := f_2(\mathcal{B}, s)$ as a function of $\Psi$. In the setting of Section 3, assume that the gradient is locally approximately constant. That is, there exists constants $K \geq 0$ such that*

$$\|\nabla^2 L_2(\Psi(\mathcal{B}^*(s), s))\| \leq K, \quad \forall s : \|s\|_1 \leq \tau.$$

*Additionally, assume that the following bilinear operator is uniformly invertible and bounded:*

$$\|(\partial_\Psi L_2^* \partial_\mathcal{B}^2 \Psi^*)^{-1}\| = \sup_{\|\mathcal{B}\|_2=1} \|(\partial_\Psi L_2^* \partial_\mathcal{B}^2 \Psi^*)^{-1}[\mathcal{B}, \cdot]\| \leq \beta.$$

*Then, for $d = 2$ (the matrix case), we derive the following closed-form, Hessian-free solution for the approximate hypergradient:*

$$G(s) := \partial_s f_1^* - diag\left[U^{*(1),\top} \partial_{U^{(1)}} f_1^* + U^{*(2),\top} \partial_{U^{(2)}} f_1^*\right] \odot s^{-1},$$

*where $\odot$ refers to the Hadamard product (entrywise).*

*Moreover, the numerical solution obtained by neglecting the Hessian satisfies the following error bound for $d = 2$:*

$$\left\|G(s) - \frac{d}{ds} f_1(\mathcal{B}^*(s), s)\right\| \lesssim K\beta.$$

*For $d > 2$ (tensor case), a tridiagonal approximation of $\partial_\mathcal{B}^2 \Psi^*$ leads to the following closed-form approximation:*

$$G(s) := \partial_s f_1^* - diag\left[\sum_{i=1}^d U^{*(i),\top} \partial_{U^{(i)}} f_1^*\right] \odot s^{-1},$$

*with the corresponding error bound:*

$$\left\|G(s) - \frac{d}{ds} f_1(\mathcal{B}^*(s), s)\right\| \lesssim K\beta + \sum_{i=1}^d \sum_{|j-i|\geq 2} \|\partial_{U^{(i)},U^{(j)}}^2 \Psi^*\|.$$

We provide a detailed proof of this theorem in Appendix B. We would like moreover to remark that, if the constraint $\mathcal{V}$ on the basis $\mathcal{B}$ is compact and $f_2$ is twice continuously differentiable, then automatically the first assumption in Theorem 4.1 is satisfied for Weierstrass theorem. This will naturally hold if we impose orthonormal constraints on the basis, as we will discuss in Section 7.

Given the approximation error in Theorem 4.1, it is important to modify the constraint in the upper-level problem to $\|s\|_1 \leq \tau + r\varepsilon = \widetilde{\tau}$ and $s \geq \varepsilon$ entrywise, where $\varepsilon > 0$ is a small constant to prevent numerical instability due to division by $s$. This adjustment ensures that the approximation remains well-behaved, particularly in cases where the singular values $s_i$ become small. More precisely, one can consider the perturbed $L^1$ simplex:

$$S := \{s \in \mathbb{R}^r \,|\, s \geq \varepsilon, \|s\|_1 \leq \widetilde{\tau}\},$$

which is still convex and thus amenable to projection-free methods, motivating our choice of stochastic Frank-Wolfe for the upper-level problem in the next Section 5.

## 5 OPTIMIZATION WITH STOCHASTIC AWAY-STEP FRANK–WOLFE

Our goal is to minimize an objective function defined as the finite sum of a set of functions. The main challenge here lies in the high computational cost: calculating the objective value or its gradient requires aggregating information across all functions in the set, which is substantially more expensive than evaluating these quantities for a single function or a bunch of them. This is especially problematic when the set of functions is large, which can quickly make optimization infeasible in practice.

In order to solve the bilevel optimization problem in Equation (1), we hence proceed as follows: first, we compute an approximate solution to the lower-level problem using a standard stochastic gradient-based method; then we compute a stochastic approximation $\widetilde{G}(s)$ of $G(s)$ by sampling a minibatch of data to represent the loss $f_1$; finally we use the $\widetilde{G}(s)$ to update the solver for the upper-level problem.

The hypergradient approximation $G(s)$ obtained in Theorem 4.1 can be efficiently computed using automatic differentiation. Specifically, to calculate $G(s)$, we first approximate the optimal basis $\mathcal{B}^*(s)$ for a fixed $s$ by optimizing for a finite number of steps, then we compute the partial derivative $\partial_s f_2(\mathcal{B}^*(s), s)$, leveraging the fact that automatic differentiation can efficiently compute $\partial_\mathcal{B} f_1^*$ in a single backpropagation step. By exploiting the structure of the parameterization, this method bypasses the need for full Hessian and allows for scalable fine-tuning even in large-scale settings.

As the upper level is a constrained optimization problem with convex simplex constraints, we approach it with a stochastic version of the away-step Frank-Wolfe algorithm. This allows us to employ a projection-free approach, which reduces the cost of the upper-level solver and allows us to obtain important theoretical guarantees of convergence to a sparse structure in a finite number of steps (See Section 6). At each iteration $s_n$, the main steps of the Frank-Wolfe algorithm (Guélat & Marcotte, 1986; Beck & Shtern, 2017; Bomze et al., 2020) are as follows:

**Frank-Wolfe direction** Compute a search direction $h_n$ that minimizes a linear local approximation of the upper-level loss function as:

$$h_n = z_n - s_n \quad \text{where} \quad z_n = \arg\min_{s \in S} \widetilde{G}(s_n)^\top s$$

Note that, as $S$ is a sparse simplex, the variable $z_n$ can be computed in an efficient way by simply looking at the entries of $\widetilde{G}(s_n)$. See Appendix C.

**Convergence criterion** Stop if $-\widetilde{G}(s_n)^\top h_n \leq \widetilde{p}$

**Away-step direction** Compute a search direction $b_n$ that maximizes a linear local approximation of the upper-level loss function as:

$$b_n = s_n - y_n, \quad \text{where} \quad y_n = \arg\max_{s \in C_n} \widetilde{G}(s_n)^\top s,$$

with $C_n = \{s \in S : \operatorname{supp}_\varepsilon(s) = \operatorname{supp}_\varepsilon(s_n)\}$ and $\operatorname{supp}_\varepsilon(s) = \{i : s_i > \varepsilon\}$. Note that this problem can be solved efficiently as for the FW direction computation by sweeping through the entries of $\widetilde{G}(s_n)$. See Appendix C.

**Steepest direction** Set $d_n = h_n$ if $-\widetilde{G}(s_n)^\top h_n \geq -\widetilde{G}(s_n)^\top b_n$. Set $d_n = b_n$, otherwise.

**Line search** Choose $\alpha_n^{\max} = 1$ if $d_n = h_n$. Otherwise, choose $\alpha_n^{\max}$ as the largest $\alpha$ such that $s_n + \alpha d_n \in S$. Then choose $\alpha_n \in (0, \alpha_n^{\max}]$ using a line search.

**Update variables** Update the current iterate as $s_{n+1} = s_n + \alpha_n d_n$. If $n \geq n_0$ then truncate $s_{n+1}$ by removing the entries that are smaller than $\varepsilon$ and remove the corresponding columns from $U$ and $V$, effectively reducing the rank of the problem.

The resulting low-rank adaptation scheme is summarized in Algorithms 1 and 2. In the next section, we will see that the proposed algorithm converges in expectation with a sublinear rate (see Theorem 6.2 for further details) and it is also able to identify the surface containing a stationary point once it gets close enough to it (see Theorem 6.5 for further details). Identifying the surface containing a solution is of paramount importance in our framework since it allows us to shrink the dimension of the problem under analysis and eventually apply more sophisticated solution techniques in the end. In Appendix G we provide a detailed per-iteration cost analysis of dEBORA, together with the real GPU consumption compared with (Zhang et al., 2023).

## 6 THEORETICAL PROPERTIES OF THE STOCHASTIC AWAY-STEP FRANK-WOLFE ALGORITHM

In this section, we propose a convergence analysis of the fine-tuning strategy. In order to simplify the notation, we let $f(s) = f_1(\mathcal{B}^*(s), s)$ where $\mathcal{B}^*(s)$ is a solution to the lower-level problem. We hence consider, in place of Problem (1), the following optimization problem:

$$\min_{s \in S} f(s), \tag{4}$$

---

**Algorithm 1** dEBORA: Efficient Bilevel Optimization-based low-Rank Adaptation

---

1: **Input:** Choose $\tau, \varepsilon > 0$, precision $\widetilde{p} > 0$, and truncation step $n_0$. Initialize adapters $\mathcal{B}$ and $s_0 \in S$
2: **For** $n = 0, 1, \ldots$
3:     Compute $\widetilde{G}(s_n)$ stochastic estimation of the hypergradient as in Algorithm 2
4:     $h_n = $ FW-direction$(\widetilde{G}(s_n), \tau, \varepsilon)$;
5:     **If** Convergence-criterion$(\widetilde{G}(s_n), h_n, \widetilde{p})$ **then** STOP
6:     $b_n = $ Away-step-direction$(\widetilde{G}(s_n), s_n, \tau, \varepsilon)$
7:     $d_n = $ Steepest-direction$\{h_n, b_n\}$
8:     $\alpha_n = $ Linesearch$(s_n, d_n)$
9:     Set $s_{n+1} = s_n + \alpha_n d_n$
10:     **If** $n \geq n_0$ **then** truncate $s_{n+1}$ by removing entries smaller than $\varepsilon$ and reduce rank accordingly
11: **End for**

---

**Algorithm 2** Stochastic approximation of the hypergradient $\widetilde{G}(s)$

---

1: **Given** $s \in S$
2: Compute an approximation $\mathcal{B}^*(s)$ to $\arg\min_{\mathcal{B}} f_2(\mathcal{B}, s)$
3: Sample a data minibatch and define $\widetilde{f}_1$ as the estimation of $f_1$ on it
4: Compute $\partial_{\mathcal{B}} \widetilde{f}_1(\mathcal{B}^*, s)$ and $\partial_s \widetilde{f}_1(\mathcal{B}^*, s)$
5: Assemble $\widetilde{G}(s) := \partial_s \widetilde{f}_1(\mathcal{B}^*, s) - \mathrm{diag}[\mathcal{B}^{*,\top} \partial_{\mathcal{B}} \widetilde{f}_1(\mathcal{B}^*, s)] \odot s^{-1}$

---

and denote with $\Delta$ the diameter of $S$. Moreover, we assume the stochastic approximation $\widetilde{G}$ of the hypergradient $\nabla f$ is available, such that the following assumption is satisfied. [1]

**Assumption 6.1.** Let us denote $e(s, \bar{s}) = (\nabla f(\bar{s}) - \widetilde{G}(\bar{s}))^\top (s - \bar{s})$. We assume that for any $\bar{s} \in S$, there exist $\chi \geq 0$ such that

$$\mathbb{E}\left[e(s, \bar{s})^2\right] \leq \chi^2 \quad \forall\, s \in S. \tag{5}$$

Remark 6.4 will discuss how the proposed analysis transfers to the chosen biased approximation $\widetilde{G}(s)$ from Algorithm 2. Notice that, by Jensen and Holder inequalities respectively, if Assumption 6.1 holds, we have the following

$$|\mathbb{E}[e(s, \bar{s})]| \leq \mathbb{E}[|e(s, \bar{s})|] \leq \mathbb{E}[(e(s, \bar{s}))^2]^{1/2} \leq \chi. \tag{6}$$

Note moreover that, in the stochastic optimization literature, it is standard to quantify the accuracy of the stochastic first-order oracle assuming bounded variance of the stochastic gradient estimates (Braun et al., 2022): for every $s \in S$, there exists $\bar{\sigma}^2 > 0$ such that $\mathbb{E}\left[\left\|\nabla f(s) - \widetilde{G}(s)\right\|^2\right] \leq \bar{\sigma}^2$. It is easy to see that this assumption on the gradient estimates implies Assumption 6.1. Indeed, we have:

$$\mathbb{E}\left[e(s, \bar{s})^2\right] \leq \mathbb{E}\left[\left\|\nabla f(\bar{s}) - \widetilde{G}(\bar{s})\right\|^2 \|s - \bar{s}\|^2\right] \leq \mathbb{E}\left[\left\|\nabla f(\bar{s}) - \widetilde{G}(\bar{s})\right\|^2\right] \|s - \bar{s}\|^2 \leq \bar{\sigma}^2 \Delta^2.$$

Let us define $g_n = -\nabla f(s_n)^\top h_n$, $\widetilde{g}_n = -\widetilde{G}(s_n)^\top h_n$ and $g_n^{FW} = -\nabla f(s_n)^\top h_n^{FW}$, with $h_n^{FW} \in \arg\min_{s \in S}\{\nabla f(s_n)^\top (s - s_n)\} - s_n$ the Frank-Wolfe direction obtained using the exact gradient. Since $S$ is a convex set, a point $s^* \in S$ is said to be stationary for the upper-level problem in (1) when $\nabla f(s^*)^\top (s - s^*) \geq 0$ for all $s \in S$. Then, $g_n^{FW}$ is an optimality measure, i.e., $g_n^{FW} = 0$ if and only if $s_n \in S$ is a stationary point. Now we show that Assumption 6.1 is satisfied with a sufficiently small $\chi$ if the stepsize $\alpha_n$ is generated with a suitable line search. Thus, Algorithm 1 converges in expectation to a stationary point at a sublinear rate on non-convex objectives with a Lipschitz continuous gradient. The constant in the convergence rate depends on the quality of the gradient estimate (the more precise the estimate, the smaller the constant). The proof can be found in Appendix D.

---

[1] We have a biased stochastic estimate of the original hypergradient, so $\mathbb{E}[\widetilde{G}] \neq \mathbb{E}[\nabla f]$.

**Theorem 6.2.** *Let $\{s_n\}$ be a sequence generated by Algorithm 1 applied to $f$ on $S$, where $\nabla f$ is Lipschitz continuous with constant $M$, $\widetilde{G}$ satisfies Assumption 6.1 with*

$$\chi \leq \frac{\eta}{2+2\eta}\widetilde{p}, \quad 0 \leq \eta < \frac{1}{3}, \tag{7}$$

*and the step size $\alpha_n$ satisfies*

$$\alpha_n \geq \bar{\alpha}_n = \min\left(\alpha_n^{max}, \frac{\widetilde{g}_n}{M\Delta^2}\right), \tag{8}$$

$$f(s_n) - f(s_n + \alpha_n d_n) \geq \rho\bar{\alpha}_n\widetilde{g}_n, \tag{9}$$

*with some fixed $\rho$ s.t.*

$$0 < \rho \leq \frac{(1-3\eta)\Delta^2 M}{2(1-\eta)^2 g_0^{FW}}. \tag{10}$$

*Then for every $T \in \mathbb{N}$,*

$$\mathbb{E}[g_T^*] \leq \sqrt{\frac{2\Delta^2 M(f(s_0) - f^*)}{T\rho(1-\eta)^2}}, \tag{11}$$

*where $g_T^* = \min\limits_{0 \leq n \leq T-1} g_n^{FW}$ and $f^* = \min_{s\in S} f(s)$.*

Equations (8) and (9) can be satisfied with suitable line searches/stepsize rules (see, e.g., (Bomze et al., 2020; 2021; Rinaldi & Zeffiro, 2022)). In particular, Lemma 6.3 below shows that this is the case for the fixed step size $\alpha_n = \bar{\alpha}_n$, and the modified Armijo line search rule which sets

$$\alpha_n = \delta^j, \tag{12}$$

where $j$ is the smallest non-negative integer such that

$$f(s_n) - f(s_n + \alpha_n d_n) \geq \gamma\alpha_n\widetilde{g}_n, \tag{13}$$

with $\gamma \in (0, 1/2)$ and $\delta \in (0, 1)$ being two fixed parameters. The proof can be found in Appendix E.

**Lemma 6.3.** *Let Assumption 6.1 hold with*

$$\chi \leq \frac{\eta}{2+2\eta}\widetilde{p}, \quad 0 \leq \eta < \frac{1}{3}. \tag{14}$$

*and let $\bar{\alpha}_n$ being defined as in Equation (8). At iteration $n$, if $\alpha_n$ is determined by*

- *the fixed step size rule*
$$\alpha_n = \bar{\alpha}_n,$$
*then Equation (8) holds and Equation (9) is satisfied with*
$$0 < \rho \leq \min\left(\frac{(1-3\eta)\Delta^2 M}{2(1-\eta)^2 g_0^{FW}}, \frac{1-\eta}{2(1+\eta)}\right).$$

- *the Armijo line search described in Equations (12) and (13), then*
$$\alpha_n \geq \min\{1, 2\delta(1-\gamma-\eta)\}\bar{\alpha}_n,$$
*and Equations (8) and (9) hold.*

*Remark* 6.4. It is easy to see that Assumption 6.1 holds with a $\chi$ that satisfies Equations (7) and (14) when the batch size chosen to build the approximate stochastic hypergradient $\widetilde{G}(s)$ is large enough. We also observe that it is possible to decompose $\chi$ in Equation (6) as a first term completely depending on the norm of the Hessian, and a second term controlled totally by the stochastic part. In particular, one would decompose the error as

$$e(s, \bar{s}) = (\nabla f(s) - G(s))^\top(s - \bar{s}) + (G(s) - \widetilde{G}(s))^\top(s - \bar{s}),$$

where $G(s)$ is again the closed form estimate of the hypergradient presented in Theorem 4.1, and $\widetilde{G}(s)$ its stochastic counterpart where the objective function is evaluated on a data batch. Under the hypothesis that the constant $\beta$ in Theorem 4.1 works across different batches, by using Cauchy-Schwarz and the triangular inequality we get

$$\mathbb{E}[|e(s, \bar{s})|] \leq K\beta\Delta + \mathbb{E}[|(G(s) - \widetilde{G}(s))^\top(s - \bar{s})|] \leq K\beta\Delta + \bar{\chi},$$

where now we define $\bar{\chi}$ as an upper bound of the third term. In particular, this allows to get potentially sharper bounds when estimating the minimal batch size needed to satisfy Equations (7) and (14).

Table 1: DeBERTaV3-base fine-tuning on GLUE benchmark.

| Method | # Params | MNLI (Acc) | SST-2 (Acc) | CoLA (Mcc) | QQP (Acc/F1) | QNLI (Acc) | RTE (Acc) | MRPC (Corr) | STS-B |
|---|---|---|---|---|---|---|---|---|---|
| Full FT | 184M | 89.90 | 95.63 | 69.19 | 92.40/89.80 | 94.03 | 83.75 | 89.46 | 91.60 |
| HAdapter | 1.22M | 90.13 | 95.53 | 68.64 | 89.27 | 94.11 | 84.48 | 89.95 | 91.48 |
| PAdapter | 1.18M | 90.33 | 95.61 | 68.77 | 89.40 | **94.29** | 85.20 | 89.46 | **91.54** |
| LoRA $r = 8$ | 1.33M | 90.29 | 95.29 | 68.57 | 90.62/90.61 | 93.91 | 85.5 | 89.75 | 89.10 |
| AdaLoRA | 1.27M | **90.44** | 95.64 | 68.76 | 90.59/90.65 | 94.11 | **86.00** | 89.44 | 91.41 |
| dEBORA $\tau = 16$ | **0.4M** | 90.01 | 95.29 | 68.72 | **91.88/89.20** | 93.42 | 83.75 | 90.16 | 90.84 |
| dEBORA (Oblique) $\tau = 16$ | 1.03M | 90.0 | 94.83 | **69.15** | 88.62/85.09 | 87.33 | 83.03 | **91.42** | 91.24 |
| dEBORA (Stiefel) $\tau = 16$ | 0.8M | 89.79 | **95.65** | 68.39 | 89.74/86.58 | 93.83 | 84.12 | 91.18 | **91.54** |

We now report a local identification result related to the Algorithm 1. More specifically, we state that our method successfully identifies the manifold containing a stationary point once it is sufficiently close to it. As we already noticed, guarantees of identification of this manifold in a finite number of steps is of critical importance within our framework, as it facilitates the reduction of the problem dimensionality and ultimately enables the application of more sophisticated solution techniques in the final stages. In other words, once the optimal face of the simplex is found, we are guaranteed that optimization can be continued just there without any repercussion, potentially drastically reducing the number of parameters.

The proof of this result can be found in Appendix F. To ease the analysis, we first introduce some useful theoretical tools and notations. We define the face of $S$ exposed by $\nabla f(s)$, with $s \in S$, as

$$\mathcal{F}_e(\nabla f(s)) = \arg\min_{z \in S} \nabla f(s)^\top z,$$

and the minimal face of $S$ containing $s \in S$ as $\mathcal{F}(s)$. We let $\lambda_v(s) = \nabla f(s)^\top (v - s)$, $v, s \in S$ and notice that for a stationary point $s^*$ of Problem (4) it holds $\lambda_v(s^*) \geq 0 \ \forall v \in V$, where $V$ is the set of extreme points in $S$. Furthermore, we define the value

$$\lambda_v^{\mathrm{MIN}}(s^*) = \min_{v \in V^+(S)} \lambda_v(s^*), \tag{15}$$

with $V^+(S) = \{v \in V : \lambda_v(s^*) > 0\}$. Finally, we indicate with $B_{\Gamma(x)}$ the ball with radius $\Gamma(x) > 0$ centered in $x$. We have:

**Theorem 6.5.** *Let $\{s_n\}$ be a sequence generated by Algorithm 1. Let $s^* \in S$ be a stationary point of Problem (4) s.t.*

- $\lambda_v^{\mathrm{MIN}}(s^*) - 2\chi > 0$,

- $\mathcal{F}_e(\nabla f(s^*)) = \mathcal{F}(s^*)$, *that is strict complementarity holds in $s^*$.*

*Then, there exists $\Gamma(s^*) > 0$ s.t. if $s_n \in B_{\Gamma(s^*)} \cap \mathcal{F}(s^*)$ then $s_{n+1} \in \mathcal{F}(s^*)$. Furthermore if the level set $\mathcal{L}(f(s_n))$ is such that $\mathcal{L}(f(s_n)) \subseteq B_{\Gamma(s^*)}$ then $s_{n+1} \in B_{\Gamma(s^*)} \cap \mathcal{F}(s^*)$.*

## 7 EXPERIMENTAL EVALUATION

In this section, we present numerical experiments to showcase the effectiveness of the proposed bilevel optimization method for fine-tuning low-rank adapters. For fair comparison, the validation set was not used in any of the two losses during training. To create the two loss functions $f_1, f_2$, we randomly partitioned the dataset into equally sized subsets, using one partition for the upper-level loss and the other for the lower-level loss. All experiments were run on a 80GB NVIDIA A100 GPU.

### 7.1 GLUE BENCHMARK

In our first experiment, we fine-tuned DeBERTaV3 (He et al., 2023) on the GLUE benchmark (Wang et al., 2019), comparing the proposed approach with state-of-the-art methods for fine-tuning large language models (LLMs). These include AdaLoRA (Zhang et al., 2023), LoRA (Hu et al., 2022), Pfeiffer adapter (Pfeiffer et al., 2021), Houlsby adapter (Houlsby et al., 2019), and full fine-tuning.

Table 2: Fine-tuning performance on tensor layers. Left: ResNet50 fine-tuning on CIFAR-10. Right: Fine-tuning of Dreambooth Stable Diffusion. First rank refers to the LoRA rank of the UNet layers, the second refers to the text encoder. In the parameter count, M stands for million and K for thousands. Initial ranks for AdaLoRA are 8 for each layer.

| Method | Val. Accuracy (%) | # Params | Method | Loss | # Params |
|---|---|---|---|---|---|
| LoRA ($r = 16$) | 89.66 | 1.17M | LoRA ($r = 2, r = 8$) | 0.245 | 3.4M |
| LoRA ($r = 8$) | 87.5 | 588K | LoRA ($r = 4, r = 8$) | 0.247 | 5.5M |
| LoRA ($r = 2$) | 70.03 | 147K | LoRA ($r = 10, r = 8$) | 0.241 | 11.3M |
| AdaLoRA | 93.28 | 316K | AdaLoRA | 0.245 | 4.7M |
| AdaLoRA | 90.97 | 79K | AdaLoRA | 0.247 | 1.78M |
| dEBORA $\tau = 8$ | **93.74** | **64K** | dEBORA $\tau = 16$ | 0.262 | 0.4M |
| dEBORA (Oblique) $\tau = 16$ | 85.65 | 66K | dEBORA (Oblique) $\tau = 16$ | **0.231** | **0.4M** |
| dEBORA (Stiefel) $\tau = 16$ | 85.83 | 72K | dEBORA (Stiefel) $\tau = 16$ | 0.252 | 0.4M |

As in (Zhang et al., 2023), the adapters were applied to each attention layer, including the query, key, value matrices, and feed-forward layers. Results are presented in Table 1.

Additionally, we tested our approach in both unconstrained and constrained settings by forcing the basis matrices to have columns of unitary norm (Oblique) or orthonormal columns (Stiefel). As both these constraints form a manifold with an explicit retraction, we compute $\mathcal{B}^*$ by performing first-order stochastic Riemannian optimization Sato (2021) at the lower level. Given the product nature of the constraints and the relatively small starting rank, retractions in these cases remained computationally affordable.

### 7.2 FINE-TUNING OF HIGHER-ORDER TENSOR LAYERS ($d > 2$)

In this experiment, we evaluate our bilevel optimizer on fine-tuning higher-order tensor layers. We tested ResNet50 (He et al., 2015) on CIFAR-10 (Krizhevsky & Hinton, 2009) and Stable Diffusion (Rombach et al., 2021), with results reported in Table 2 (left) and Table 2 (right), respectively.
For ResNet50, we applied adapters to each convolutional layer while retraining the final fully connected layer with an appropriately sized one. All methods used the same training settings: a constant learning rate of $5 \times 10^{-1}$, weight decay of $1 \times 10^{-3}$, LoRA $\alpha = 32$, and no dropout. We note that the original LoRA implementation (Hu et al., 2022) for convolutional layers uses matrix factorization of the weight tensor, scaling the number of parameters as $O(r(F + Ck^2))$. In contrast, our CP-like factorization scales more efficiently as $O(r(2k + C + F))$, where $F$ is the number of output features, $C$ is the number of input channels, and $k$ is the kernel size. This more efficient representation is reflected in the total number of trainable parameters reported in Table 2 (left). Despite using fewer parameters, our approach competes well with the baselines, particularly in the unconstrained setting, which is computationally less challenging due to the absence of retractions.

For the Stable Diffusion experiment, we inserted adapters into the convolutional layers of the UNet and the linear layers of the text encoder. All models were trained under the same conditions as in (Mangrulkar et al., 2022), and the results are reported in Table 2 (right). For AdaLoRA, we implemented the same tensor-based convolutional adapter representation used in our method, which led to higher compression compared to LoRA, which does not utilize tensor factorization. Even with a similar parameterization, our method consistently outperforms the other baselines, achieving better results with fewer effective parameters.

## 8 CONCLUSION

In this paper, we introduced a novel bilevel optimization framework for low-rank adaptation, providing a parameter-efficient fine-tuning strategy for large-scale neural networks. Our approach leverages a dynamic rank selection mechanism within a bilevel structure, enabling efficient adaptation while minimizing the number of trainable parameters. A key contribution of our work is the theoretical analysis of the proposed stochastic bilevel optimizer, where we established convergence guarantees. Through a variety of experiments on the GLUE benchmark, ResNet50, and Stable Diffusion, we demonstrated that our method consistently matches or outperforms existing state-of-the-art approaches such as AdaLoRA and LoRA, while significantly reducing the parameter count.

## FUNDING ACKNOLEDGEMENTS

The work of Francesco Rinaldi has been partially funded by the EuropeanUnion - NextGenerationEU under the National Recovery and Resilience Plan (NRRP), Mission 4 Component 2 Investment 1.1 - Call PRIN 2022 No. 104 of February 2, 2022 of Italian Ministry of University and Research; Project 2022BMBW2A (subject area: PE - Physical Sciences and Engineering) "Large-scale optimization for sustainable and resilient energy systems.", CUP I53D23002310006.

The work of E. Zangrando was funded by the MUR-PNRR project "Low-parametric machine learning". Francesco Tudisco is partially funded by the PRIN-MUR project MOLE code: 2022ZK5ME7 MUR D.D. financing decree n. 20428 of November 6th, 2024, CUP B53C24006410006; and by the PRIN-PNRR project FIN4GEO within the European Union's Next Generation EU framework, Mission 4, Component 2, CUP P2022BNB97.

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

## A   APPENDIX

In the next sections, we will include the proofs of the results in the main text and a schematic implementation of the training procedure.

## B    PROOF OF THEOREM 4.1

The hypergradient equation given in Equation (2) requires just to calculate $\partial_s \mathcal{B}^*(s)$, i.e., a solution of the implicit gradient Equation (3)

$$\partial_{\mathcal{B}}^2 f_2^* \partial_s \mathcal{B}^* + \partial_s \partial_{\mathcal{B}} f_2^* = 0. \tag{16}$$

Using the chain rule and the fact that $f_2(\mathcal{B}, s) = L_2(\Psi(\mathcal{B}, s))$, we have that

$$\partial_{\mathcal{B}}^2 f_2^* = \partial_{\mathcal{B}} \partial_{\mathcal{B}} f_2^* = \partial_{\mathcal{B}} \partial_{\Psi} L_2^* \partial_{\mathcal{B}} \Psi^* = \partial_{\Psi}^2 L_2^* \partial_{\mathcal{B}} \Psi^* \partial_{\mathcal{B}} \Psi^* + \partial_{\Psi} L_2^* \partial_{\mathcal{B}}^2 \Psi^*. \tag{17}$$

In a similar manner, we have that

$$\partial_s \partial_{\mathcal{B}} f_2^* = \partial_{\mathcal{B}} \partial_{\Psi} L_2^* \partial_s \Psi^* = \partial_{\Psi}^2 L_2^* \partial_{\mathcal{B}} \Psi^* \partial_s \Psi^* + \partial_{\Psi} L_2^* \partial_{\mathcal{B}} \partial_s \Psi^*. \tag{18}$$

By plugging Equations (17) and (18) in equation Equation (16) we get the following:

$$(\partial_{\Psi}^2 L_2^* \partial_{\mathcal{B}} \Psi^* \partial_{\mathcal{B}} \Psi^* + \partial_{\Psi} L_2^* \partial_{\mathcal{B}}^2 \Psi^*) \partial_s \mathcal{B}^* + \partial_{\Psi}^2 L_2^* \partial_{\mathcal{B}} \Psi^* \partial_s \Psi^* + \partial_{\Psi} L_2^* \partial_{\mathcal{B}} \partial_s \Psi^* = 0,$$

by suitably rearranging this expression, we have

$$\partial_{\Psi}^2 L_2^* \partial_{\mathcal{B}} \Psi^* (\partial_{\mathcal{B}} \Psi^* \partial_s \mathcal{B}^* + \partial_s \Psi^*) + \partial_{\Psi} L_2^* (\partial_{\mathcal{B}}^2 \Psi^* \partial_s \mathcal{B}^* + \partial_s \partial_{\mathcal{B}} \Psi^*) = 0. \tag{19}$$

Assuming for the moment that $\partial_{\Psi}^2 L_2^* \equiv 0$, then the solution of Equation (19) is given by the solution of

$$\partial_{\mathcal{B}}^2 \Psi^* \partial_s \mathcal{B}^* = -\partial_s \partial_{\mathcal{B}} \Psi^*.$$

Since $\Psi(\mathcal{B}, s)$ is multilinear in the basis matrices $\mathcal{B} = (U^{(1)}, \dots, U^{(d)})$, the partial derivatives have diagonal terms equal to zero, i.e. $\partial_{U^{(i)}}^2 \Psi^* = 0$, we get the following system of equations

$$\sum_{j=1, j \neq i}^{d} \partial_{U^{(i)}, U^{(j)}}^2 \Psi^* \partial_s U^{(j)} = -\partial_s \partial_{U^{(i)}} \Psi^*, \quad i = 1, \dots, d.$$

We hence just need to calculate

$$\partial_{U^{(i)}, U^{(j)}}^2 \Psi^* = \partial_{U^{(i)}, U^{(j)}} \sum_{k=1}^{r} s_k U^{(1)} e_k \otimes \cdots \otimes U^{(i)} e_k \otimes \dots U^{(j)} e_k \otimes \cdots \otimes U^{(d)} e_k.$$

For $d \geq 3$, the last system cannot be solved in closed form, but given the symmetry of the Hessian one could exploit the structure through the use of Krylov methods. For $d = 2$ instead, the last calculation leads to the following equality:

$$s_\beta \sum_\gamma \partial_{s_\eta} U_{\gamma\beta}^{(2)} e_\alpha \otimes e_\gamma = -\delta_{\beta\eta} e_\alpha \otimes U^{(2)} e_\beta.$$

By testing the last equation on the left by $e_i$ and on the right by $e_j$ we get:

$$s_\beta \delta_{\alpha i} \partial_{s_\eta} U_{j\beta}^{(2)} = -\delta_{\beta\eta} \delta_{\alpha i} U_{j\beta}^{(2)},$$

that finally leads to

$$\partial_{s_\eta} U_{j\beta}^{(2)} = -\delta_{\beta\eta} U_{j\beta}^{(2)} s_\beta^{-1},$$

$$\partial_{s_\eta} U_{j\beta}^{(1)} = -\delta_{\beta\eta} U_{j\beta}^{(1)} s_\beta^{-1}.$$

We hence get the final hypergradient expression by means of Equation (2):

$$\frac{d}{ds} f_1(\mathcal{B}^*(s), s) = \partial_s f_1^* - diag[U^{(1)*, \top} \partial_{U^{(1)}} f_1^* + U^{(2)*, \top} \partial_{U^{(2)}} f_1^*] \odot s^{-1}.$$

We are just left to estimate the error between this approximate solution and the real hypergradient. The structure of Equation (19) can be summarized by giving an error estimate for a linear system of the form

$$A(Bx + b) + C(Dx + d) = 0,$$
$$A = \partial_{\Psi}^2 L_2^*, B = \partial_{\mathcal{B}} \Psi^* \partial_{\mathcal{B}} \Psi^*,$$
$$C = \partial_{\Psi} L_2^*, D = \partial_{\mathcal{B}}^2 \Psi^*,$$
$$b = \partial_s \Psi^*, d = \partial_s \partial_{\mathcal{B}} \Psi^*,$$
$$x = \partial_s \mathcal{B}^*.$$

Now let $x^* = -(AB + CD)^{-1}(Ab - Cd)$ be the solution of the above linear system, and let $y^* = -D^{-1}d$ the solution of $Dx + d = 0$, which exactly represents our approximation. Let us define $M = AB + CD$ and estimate the distance between the two solutions:

$$\|y^* - x^*\| = \left\|\left(D^{-1} - M^{-1}C\right)d + M^{-1}Ab\right\| \leq \|D^{-1} - M^{-1}C\|\|d\| + \|M^{-1}\|\|A\|\|b\|.$$

By using the fact that for two invertible linear operators $P, Q$, the identity $(P + Q)^{-1} = P^{-1} - P^{-1}Q(P + Q)^{-1}$ holds, we get that

$$D^{-1} - M^{-1}C = D^{-1} - ((CD)^{-1} - (CD)^{-1}(AB)M^{-1})C = (CD)^{-1}ABM^{-1}C. \quad (20)$$

By inserting Equation (20) in the last estimate we get

$$\|y^* - x^*\| \leq \|(CD)^{-1}\|\|A\|\|BM^{-1}C\|\|d\| + \|M^{-1}\|\|A\|\|b\| = \alpha\|A\|,$$
$$\alpha = \|(CD)^{-1}\|\|BM^{-1}C\|\|d\| + \|M^{-1}\|\|b\|,$$

from which, by using the hypothesis on the uniform invertibility and boundedness of $CD$, and the control on the Hessian norm, we get by the use of Neumann expansion for $\|AB(CD)^{-1}\| \leq K\beta < 1$

$$\|M^{-1}\| \leq \frac{\|(CD)^{-1}\|}{1 - \|(CD)^{-1}\|\|AB\|} \leq \frac{\beta}{1 - K\beta\|B\|}.$$

Finally, we get

$$\|y^* - x^*\| \leq \left(\frac{\beta^2}{1 - K\beta\|B\|}\|B\|\|C\|\|d\| + \frac{\beta}{1 - K\beta\|B\|}\|b\|\right)\|A\| \lesssim \beta K,$$

where the omitted constant is bounded because the basis lies in a product of Stiefel manifolds, which are compact, and thus all the terms in parenthesis can be bounded.

For the general tensor case $d > 2$, following the previous calculation, a tridiagonal approximation $\widetilde{D}$ of $D$ immediately leads to an error bound:

$$\|y^* - x^*\| \lesssim \beta K + \|D^{-1} - \widetilde{D}^{-1}\| \leq \beta K + \|D^{-1}\|\|\widetilde{D}^{-1}\|\|D - \widetilde{D}\|.$$

By taking the difference

$$\left\|G(s) - \frac{d}{ds}f_1^*(s)\right\| \leq \|\partial_{\mathcal{B}}f_1^*\|\|x^* - y^*\|,$$

and using the boundedness of $\|\partial_{\mathcal{B}}f_1^*\|$, we get the desired result.

## C  ADDITIONAL DETAILS FOR SECTION 5

In this section, we report additional details on the computational cost of the Stochastic Away-Step Frank–Wolfe algorithm presented in Section 5.

The optimization problems used to determine the Frank-Wolfe and Away-step directions can be solved efficiently since we are concentrating on minimizing a function over the perturbed $L^1$ simplex. In the Frank-Wolfe step, we indeed minimize a linear function over a polytope $S$. We know, by means of the fundamental theorem of linear programming, that one of the vertices of $S$ is solution of such a linear program (See,e.g., (Bertsekas, 2016)). Therefore we simply need to evaluate that function on this finite set of elements, a task that can be performed at a linear cost. In particular,

$$G(s_n)^\top v_i = \begin{cases} \varepsilon \sum_{j=1}^r G(s_n)_j & \text{if } i = 0, \\ \varepsilon \sum_{j=1}^r G(s_n)_j + \tau G(s_n)_i & \text{otherwise.} \end{cases}$$

with $V = \{v_0, v_1, \ldots, v_r\}$ the set of vertices of our feasible set $S$.

Therefore, if $G(s_n)_i > 0 \quad \forall i = 1, \ldots, r$ then $\hat{i} = 0$, otherwise we choose $\hat{i} \in \arg\min_i G(s_n)_i$.

Similarly, in the Away step, we simply need to evaluate the function on the set of active vertices $S_n$ used to describe the iterate. Therefore, if $0 \in S_n$ and $G(s_n)_j < 0 \quad \forall j \in S_n$ then $\hat{j} = 0$, otherwise we choose $\hat{j} \in \arg\max_{j \in S_n} G(s_n)_j$.

## D    PROOF OF THEOREM 6.2

We first notice that, using barycentric coordinates, Problem (4) can be written in the following form:

$$
\begin{aligned}
\min \quad & f(s) \\
\text{s.t.} \quad & s = Aw \\
& e^\top w = 1 \\
& w \geq 0\,,
\end{aligned}
$$

with $A = [v_0, v_1, \ldots, v_r] \in \mathbb{R}^{r \times (r+1)}$. So given a feasible iterate $s_n \in \mathbb{R}^r$ in Problem (4), we can suitably define the non-negative weights $w \in \mathbb{R}^{r+1}$ related to the barycentric coordinates and define the set $S_n = \{j : w_j > 0\}$, the set of indices related to the so-called active vertices, that is the vertices with a positive weight $w_j$ in the barycentric coordinates. It is then easy to see that, in this case, the linearized problem defined in the Away-step-direction procedure (Step 6 of our algorithm), simply reduces to select the vertex $v_{\hat{\jmath}}$ that maximizes the scalar product $\widetilde{G}(s_n)^\top v_{\hat{\jmath}}$, with $\hat{\jmath} \in S_n = \{j : w_j > 0\}$.

The following chain of inequalities holds:

$$
-\nabla f(s_n)^\top h_n^{FW} \geq -\nabla f(s_n)^\top h_n \geq -\widetilde{G}(s_n)^\top h_n - \epsilon \geq -\widetilde{G}(s_n)^\top h_n^{FW} - \epsilon \geq -\nabla f(s_n)^\top h_n^{FW} - 2\epsilon, \quad (21)
$$

where we used Equations (5) and (6) in the second and the last inequality with $\epsilon = \chi + \chi = 2\chi$, while the first and the third inequality follow from the definition of $h_n^{FW}$ and $h_n$.
In particular, using the definitions of $\widetilde{g}^n$, $g_n$ and $g_n^{FW}$, from Equation (21) we can write

$$
g_n^{FW} \geq \widetilde{g}_n - \epsilon, \tag{22}
$$

$$
\widetilde{g}_n \geq g_n^{FW} - \epsilon, \tag{23}
$$

$$
g_n \geq \widetilde{g}_n - \epsilon. \tag{24}
$$

Using Equations (7) and (22), we also have

$$
\epsilon \leq \eta(\widetilde{g}_n - \epsilon) \leq \eta g_n^{FW}. \tag{25}
$$

Now, let us distinguish three cases.

**A)** If $\bar{\alpha}_n < \alpha_n^{max}$, from Equation (8) it follows that $\dfrac{\widetilde{g}_n}{M\Delta^2} < \alpha_n^{max}$ and $\bar{\alpha}_n = \dfrac{\widetilde{g}_n}{M\Delta^2}$. Let $\mathbb{1}_A$ denote the indicator function for this case. We observe that, from Equations (23) and (25), we have

$$
\widetilde{g}_n \geq (1 - \eta)g_n^{FW}. \tag{26}
$$

Using Equations (9) and (26), we can write

$$
\mathbb{1}_A\{f(s_n) - f(s_n + \alpha_n d_n)\} \geq \mathbb{1}_A\{\rho\bar{\alpha}_n\widetilde{g}_n\} \geq \mathbb{1}_A\left\{\frac{\rho\widetilde{g}_n^2}{M\Delta^2}\right\} \geq \mathbb{1}_A\left\{\frac{\rho(1-\eta)^2}{M\Delta^2}(g_n^{FW})^2\right\}. \tag{27}
$$

As for $S_n$, by hypothesis, we have either $d_n = h_n = v_{\hat{\imath}} - s_n$ or $d_n = b_n = s_n - v_{\hat{\jmath}}$ for some $\hat{\imath}, \hat{\jmath} \in [0 : r]$. In particular, $S_{n+1} \subseteq S_n \cup \{\hat{\imath}\}$ so that $|S_{n+1}| \leq |S_n| + 1$.

**B)** If $\alpha_n = \bar{\alpha}_n = \alpha_n^{max}$ and $d_n = h_n$, then $\bar{\alpha}_n = \alpha_n^{max} = 1$. Let $\mathbb{1}_B$ denote the indicator function for this case. By the standard descent lemma, we can write

$$
\begin{aligned}
\mathbb{1}_B\{f(s_n) - f(s_{n+1})\} \;&=\; \mathbb{1}_B\{f(s_n) - f(s_n + d_n)\} \geq \\
&\geq\; \mathbb{1}_B\left\{g_n - \frac{M}{2}\|d_n\|^2\right\} \geq \mathbb{1}_B\left\{g_n - \frac{M\Delta^2}{2}\right\}.
\end{aligned}
$$

Using Equation (24) and the fact that $\frac{\widetilde{g}_n}{M\Delta^2} \geq 1$, we obtain

$$
\mathbb{1}_B\{f(s_n) - f(s_{n+1})\} \geq \mathbb{1}_B\left\{\widetilde{g}_n - \epsilon - \frac{M\Delta^2}{2}\right\} \geq \mathbb{1}_B\left\{\frac{\widetilde{g}_n}{2} - \epsilon\right\}.
$$

From Equations (23) and (25), we also have

$$\frac{\widetilde{g}_n}{2} - \epsilon \geq \frac{g_n^{FW}}{2} - \frac{3}{2}\epsilon \geq \frac{g_n^{FW}}{2} - \frac{3}{2}\eta g_n^{FW} = \frac{1 - 3\eta}{2} g_n^{FW}.$$

Hence, we can write

$$\mathbb{1}_B\{f(s_n) - f(s_{n+1})\} \geq \mathbb{1}_B \left\{ \frac{1 - 3\eta}{2} g_n^{FW} \right\} \geq \mathbb{1}_B \left\{ \frac{\rho(1 - \eta)^2}{M\Delta^2}(g_n^{FW})^2 \right\}, \qquad (28)$$

where in the last inequality we used Equation (10).

Reasoning as in the case above we also have $|S_{n+1}| \leq |S_n| + 1$.

**C)** If $\alpha_n = \bar{\alpha}_n = \alpha_n^{max}$ and $d_n = b_n = s_n - v_{\hat{j}}$ for $\hat{j} \in S_n$. Then we have $w_n^i = 0$ for $i \in [1:r] \setminus S_n \cup \{\hat{j}\}$ and $w_{n+1}^i > 0$ for $i \in S_n \setminus \{\hat{j}\}$. In particular, $|S_{n+1}| = |S_n| - 1$.

Now, based on the three cases analyzed above, we partition the iterations $\{0, 1, \ldots, T - 1\}$ into three subsets $N_A, N_B, N_C$ and defined as follows:

$$N_A(T) = \{n < T \colon \bar{\alpha}_n < \alpha_n^{max}\},$$
$$N_V(T) = \{n < T \colon \bar{\alpha}_n = \alpha_n^{max}, d_n = h_n\},$$
$$N_C(T) = \{n < T \colon \bar{\alpha}_n = \alpha_n^{max}, d_n = b_n\}.$$

We have by induction on the recurrence relation proved for $|S_n|$ that

$$|S_T| - |S_0| \leq |N_A(T)| + |N_B(T)| - |N_C(T)| \quad \text{for every } T \in \mathbb{N}. \qquad (29)$$

Since $N_C(T) = T - |N_A(T)| - |N_B(T)|$, from Equation (29) we get

$$|N_A(T)| + |N_B(T)| \geq \frac{T + |S_T| - |S_0|}{2} \geq \frac{T}{2}, \qquad (30)$$

where we used $|S_0| = \ell \leq |S_T|$.

Considering just the good case, where we can have a bound on the decrease, using Equations (27) and (28), we can write

$$\mathbb{E}_n\left[f(s_n) - f(s_{n+1})\right] = \mathbb{E}_n\left[(\mathbb{1}_A + \mathbb{1}_B)\{f(s_n) - f(s_{n+1})\}\right] \geq \frac{\rho(1 - \eta)^2}{M\Delta^2}(g_n^{FW})^2. \qquad (31)$$

Using Equation (31) and taking full expectations, we obtain

$$
\begin{aligned}
f(s_0) - f^* &\geq \sum_{n=0}^{T-1} \mathbb{E}\left[f(s_n) - f(s_{n+1})\right] \\
&\geq \sum_{N_A(T)} \mathbb{E}\left[f(s_n) - f(s_{n+1})\right] + \sum_{N_B(T)} \mathbb{E}\left[f(s_n) - f(s_{n+1})\right] \\
&\geq |N_A(T)| \min_{n \in N_A(T)} \frac{\rho(1 - \eta)^2}{M\Delta^2} \mathbb{E}\left[(g_n^{FW})^2\right] + |N_B(T)| \min_{n \in N_B(T)} \frac{\rho(1 - \eta)^2}{M\Delta^2} \mathbb{E}\left[(g_n^{FW})^2\right] \\
&\geq |N_A(T)| \min_{n \in N_A(T)} \frac{\rho(1 - \eta)^2}{M\Delta^2} \mathbb{E}\left[g_n^{FW}\right]^2 + |N_B(T)| \min_{n \in N_B(T)} \frac{\rho(1 - \eta)^2}{M\Delta^2} \mathbb{E}\left[g_n^{FW}\right]^2 \\
&\geq (|N_A(T)| + |N_B(T)|)\frac{\rho(1 - \eta)^2}{M\Delta^2} \min_{0 \leq n \leq T-1} \mathbb{E}\left[g_n^{FW}\right]^2 \\
&\geq (|N_A(T)| + |N_B(T)|)\frac{\rho(1 - \eta)^2}{M\Delta^2} \mathbb{E}\left[g_T^*\right]^2 \\
&\geq \frac{T}{2} \frac{\rho(1 - \eta)^2}{M\Delta^2} \mathbb{E}\left[g_T^*\right]^2,
\end{aligned}
$$

where the fourth inequality follows from Jensen's formula, in the second-last inequality we use the definition of $g_T^*$ and the last inequality is due to Equation (30).

Hence,

$$\mathbb{E}\left[g_T^*\right] \leq \sqrt{\frac{2\Delta^2 M(f(s_0) - f^*)}{T\rho(1-\eta)^2}},$$

leading to the desired result.

## E   PROOF OF LEMMA 6.3

The proof partially follows the one in (Venturini et al., 2023). Reasoning as in the proof of Theorem 6.2, we have that Equation (24) holds. By the standard descent lemma, we can write

$$f(s_n) - f(s_n + \alpha d_n) \geq \alpha g_n - \alpha^2 \frac{M\|d_n\|^2}{2} \geq \alpha(\widetilde{g}_n - \epsilon) - \alpha^2 \frac{M\Delta^2}{2}, \quad \forall \alpha \in \mathbb{R}, \quad (32)$$

where the last inequality follows from Equation (24).
At iteration $n$, if $\alpha$ is determined by:

- $\alpha = \bar{\alpha}_n$, then we can replace the stepsize in the right-hand side with $\frac{\widetilde{g}_n}{M\Delta^2}$ in Equation (32), obtaining

$$f(s_n) - f(s_n + \bar{\alpha}_n d_n) \geq \frac{\widetilde{g}_n(\widetilde{g}_n - \epsilon)}{M\Delta^2} - \frac{\widetilde{g}_n^2}{2M\Delta^2} = \frac{\widetilde{g}_n^2}{2M\Delta^2} - \frac{\widetilde{g}_n \epsilon}{M\Delta^2}.$$

  Using Equation (14), we obtain

$$f(s_n) - f(s_n + \bar{\alpha}_n d_n) \geq \frac{\widetilde{g}_n^2}{2M\Delta^2} - \frac{1}{M\Delta^2}\widetilde{g}_n^2\left(\frac{\eta}{1+\eta}\right) = \frac{1-\eta}{2(1+\eta)}\frac{\widetilde{g}_n^2}{M\Delta^2}.$$

  Thus, we have

$$f(s_n) - f(s_n + \alpha_n d_n) \geq \rho\frac{\widetilde{g}_n^2}{M\Delta^2} \geq \rho\bar{\alpha}_n\widetilde{g}_n.$$

- the Armijo line search, then

$$f(s_n) - f(s_n + \alpha d_n) \geq \gamma\alpha\widetilde{g}_n \quad \forall \alpha \in \left[0, 2\frac{(1-\gamma)\widetilde{g}_n - \epsilon}{M\Delta^2}\right].$$

  Since $\alpha_n$ is computed by Equations (12) and (13), we can write

$$\alpha_n \geq \min\left(1, 2\delta\frac{(1-\gamma)\widetilde{g}_n - \epsilon}{M\Delta^2}\right)$$

$$\geq \min\left(1, 2\delta\frac{(1-\gamma-\eta)\widetilde{g}_n}{M\Delta^2}\right)$$

$$\geq \min(1, 2\delta(1-\gamma-\eta))\bar{\alpha}_n,$$

  where the second inequality follows from Equation (14).
  We hence have

$$\alpha_n \geq \min\left(1, c\frac{\widetilde{g}_n}{M\Delta^2}\right) \text{ for some } c > 0.$$

  We now consider two cases: when $c \geq 1$ then $\bar{\alpha}_n$ is of course a lower bound for the step size $\alpha_n$, and when $c < 1$ we can still recover Equation (8) by considering $\widetilde{M} = M/c$ instead of $M$ as Lipschitz constant.

## F   PROOF OF THEOREM 6.5

By the stationarity of $s^*$ and by definition of exposed face, we can write

$$\lambda_v(s^*) \geq 0 \quad \forall v \in V, \quad (33)$$

$$\lambda_v(s^*) = 0 \quad iff \quad v \in \mathcal{F}(s^*). \quad (34)$$

By continuity, we can choose

$$\Gamma(s^*) = \frac{\lambda_v^{MIN}(s^*) - 2\chi}{2} > 0 \quad s.t. \tag{35}$$

$$\lambda_v(s_n) > \lambda_v(s^*) - \Gamma(s^*) \quad \forall v \in V \setminus (V \cap \mathcal{F}(s^*)). \tag{36}$$

Using Equations (5) and (6) and Equation (36), we can write

$$\widetilde{G}(s_n)^\top(v - s_n) + 2\chi \geq \nabla f(s_n)^\top(v - s_n) > \lambda_v(s^*) - \Gamma(s^*). \tag{37}$$

Therefore,

$$\widetilde{G}(s_n)^\top(v - s_n) \geq \lambda_v(s^*) - 2\chi - \frac{\lambda_v^{MIN}(s^*) - 2\chi}{2} > 0 \quad \forall v \in V \setminus (V \cap \mathcal{F}(s^*)),$$

where in the first inequality we used Equation (37) and in the second inequality we used Equation (34) and the definitions of $\lambda_v^{MIN}(s^*)$ in Equation (15) and $\Gamma(s^*)$ in Equation (35).

The algorithm will then only choose a Frank-Wolfe or an away direction that guarantees the new iterate $s_{n+1}$ to stay in the face, so $s_{n+1} \in \mathcal{F}(s^*)$. If we further have that the level set $\mathcal{L}(f(s_n))$ is such that $\mathcal{L}(f(s_n)) \subset B_{\Gamma(s^*)}$ then, considering the fact that the sequence $\{f(s_n)\}$ is non-increasing, we can easily see that $s_{n+1} \in B_{\Gamma(s^*)} \cap \mathcal{F}(s^*)$.

## G   PER-ITERATION COST OF DEBORA

We can give an upper bound on the number of arithmetic operations carried out at every dEBORA iteration: each iteration indeed requires $T_{max}C_r$ operations, where $C_r$ is the cost of one lower-level optimization step with rank $r$ and $T_{max}$ is the maximal number of iterations for the lower-level optimization step. Notice that $C_r$ includes the cost of forward, backward, and the optimization step. In particular, $C_r$ is equal to the cost of performing one optimization step for (Zhang et al., 2023) without rank adaptation. Moreover, we need to add the cost of calculating the hypergradient which is $\mathcal{O}(Lrn^2)$, where $L$ is the number of layers in the neural network (assuming for simplicity layers have square dimensions of dimension $n$).

Notice that when the correct face is identified, the cost of calculating the single iteration at the lower-level $C_r$ and the cost of the upper-level step decreases. Finally, we have a linear minimization oracle cost of $\mathcal{O}(r)$, and an optional line search with cost $\mathcal{O}(1)$. The cost of calculating the Frank-Wolfe direction (Linear minimization oracle) is $\mathcal{O}(r)$ because it requires solving the subproblem $\min_{s \in S} \widetilde{G}(s_n)^\top s$. This is a linear minimization problem on a convex polytope, so its solution lies in one of the vertices for the fundamental theorem of linear programming: since vertices are the null vector and the vectors of the canonical basis, the solution can be found by simply looking at the entries of $\widetilde{G}(s_n)$ and finding the smallest one.

Summing up, each iteration of the method has a cost of $\mathcal{O}(T_{max}C_r + Lrn^2)$ against a $\mathcal{O}(C_r)$ for (Hu et al., 2022) and a $\mathcal{O}(C_r + Lrn^2)$ for (Zhang et al., 2023) (including their cost of calculating the sensitivity metric for rank truncation). While the single iteration of dEBORA is more expensive than the one in (Zhang et al., 2023) by a factor $T_{max}$ (which can be predefined by the user as the maximal number of iterations for the lower-level), in Figure 4 we report different GPU statistics during training on the MRPC task (GLUE benchmark) showing that the effective computing time of dEBORA is competitive. As we can observe from Figure 4, the effective memory consumption is lower together with the average GPU power usage. Moreover, while the cost of single iteration is bigger, the effective time to same accuracy is comparable.

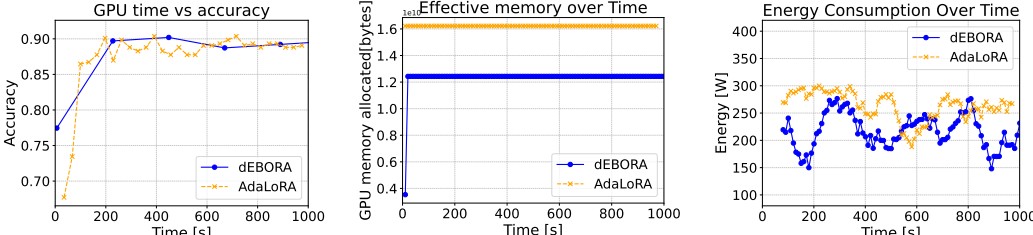

Figure 1: Time vs accuracy for the MRPC task.

Figure 2: GPU memory consumption for the MRPC task.

Figure 3: Energy consumption on the MRPC task.

Figure 4: GPU statistics during training (A100 80GB).

