# OpenReview forum: "dEBORA: Efficient Bilevel Optimization-based low-Rank Adaptation"
_ICLR.cc/2025/Conference — ICLR 2025 Poster_

### Official Review · Reviewer_1wqr · 2024-10-31

**Soundness:** 3
**Presentation:** 3
**Contribution:** 3
**Rating:** 8
**Confidence:** 3

**Summary:**

This work introduces a compact bilevel optimization framework for low-rank adaptation, offering an efficient fine-tuning strategy for large-scale neural networks via dynamic rank selection. Theoretical analysis provides convergence guarantees, and numerical experiments demonstrate that proposed method achieves performance comparable to advanced approaches.

**Strengths:**

- A relatively fresh application of the Frank-Wolf method.

**Weaknesses:**

See questions below.

**Questions:**

- Line 150-151, maybe it's better to use a figure to visualize the bypass in the neural networks.\
- Line 159, what's the intuition of splitting dataset to two parts?\
- Line 249, I understand you are focusing on the (stochastic) Frank-Wolf with mini batch of data, have you considered the full batch size?\
- Line 472, "randomly partitioned the dataset into equally sized subsets", did you ensure the label balance for both dataset?

---

> ### Author Response · Authors · 2024-11-15
>
> First of all, we would like to thank the reviewer for all the comments. Below, we provide a point-by-point response to questions raised.
>
> 1. That is a nice idea, we are working on an illustration to add.
> 2. Splitting the dataset in two parts is not necessary, it is possible to use the full training dataset on both levels of the problem ($\mathcal D_1 = \mathcal D_2 = \mathcal D$). It just makes the formulation more general, and it allows us to use a smaller dataset on the lower level, making the hypergradient computation cheaper.
> 3. Our goal is to minimize an objective function defined as the finite sum of a set of functions. The main challenge here lies in the high computational cost: calculating the objective value or its gradient requires aggregating information across all functions in the set, which is substantially more expensive than evaluating these quantities for a single function or a bunch of them (like in our Stochastic Frank-Wolfe approach). This is especially problematic when the set of functions is large, as each computation scales with the number of functions, quickly making optimization infeasible in practice. This is the main reason why also Stochastic Gradient fits better than Batch Gradient when dealing with Machine/Deep  Learning applications. We now highlight this at the beginning of Section 5 of the revised manuscript.
> 4. Yes, label balance was ensured.

---

> > ### Comment · Reviewer_1wqr · 2024-11-27
> >
> > Thank you for your patient response and addressing my concerns.

---

### Official Review · Reviewer_AG3z · 2024-11-01

**Soundness:** 3
**Presentation:** 3
**Contribution:** 3
**Rating:** 6
**Confidence:** 4

**Summary:**

This paper tackles the problem of optimal rank selection in low-rank adaptation methods, which is essential for efficient fine-tuning of large-scale neural networks. The authors propose a bilevel optimization strategy that trains matrix and tensor low-rank adapters while dynamically selecting the best rank per layer. Their approach avoids implicit differentiation, uses a stochastic variant of the Frank-Wolfe algorithm, and provides identifiability guarantees for the optimal rank structure. Theoretical analysis and numerical experiments support the method's efficiency and effectiveness, showcasing adaptive parameter allocation across network layers.

**Strengths:**

1.	This paper addresses a longstanding issue in LoRA: selecting the optimal rank for each low-rank adapter, a key factor in balancing model performance and computational efficiency. Whereas traditional PEFT algorithms require manual tuning of this parameter, the authors propose a criterion based on optimization.
2.	To address this issue, it introduces a bilinear optimization approach that alternates between optimizing the LoRA matrices and their ranks, presenting a novel contribution to the PEFT community.
3.	With fewer parameters, the proposed method demonstrates competitive or superior performance compared to existing approaches such as LoRA and AdaLoRA.

**Weaknesses:**

[key issue] While selecting the optimal rank is indeed critical for LoRA, in many practical applications, concerns about memory consumption and training time are often more pressing. The proposed bilinear optimization approach suggests a reduction in parameter count, but the practical impact on memory usage remains uncertain. A detailed comparison of memory consumption in real-world scenarios would strengthen the paper’s contribution.

[key issue] The proposed method has been evaluated on architectures such as DeBERTa and ResNet. However, the reviewer is interested in its applicability to even larger models. Specifically, would this bilevel optimization approach introduce significant additional computation time? It would be beneficial if the authors could provide both theoretical analysis and empirical results addressing this question. For instance, would the use of a Frank-Wolfe optimization strategy substantially increase training time on large-scale models such as LLaMA2?

In Theorem 4.1, the condition K≥0 alone appears insufficient to ensure that the gradient is locally constant. Additionally, the assumptions regarding the uniform invertibility and boundedness of the bilinear operator merit further examination for practical feasibility.
The parameter τ is tied to the rank selection process, yet its practical setting and impact on memory usage are not fully clarified.

Could the authors discuss guidelines for selecting τ in practice and elaborate on its relationship with memory requirements?

In line 154, should the parameter s be restricted to nonnegative values? Further clarification on this point would be helpful.

**Questions:**

See the above "weakness".

---

> ### Author Response · Authors · 2024-11-15
>
> We wish first of all to thank the reviewer for all the comments and questions. Below, we provide a point-by-point response to questions raised. Moreover, we uploaded a revised version of the manuscript including additional results and some additional clarification.
>
> 1. Thank you for pointing out this unclear aspect of memory. Could you please clarify what type of memory consumption you are suggesting? In theory, our method requires exactly the same memory consumption as AdaLoRA. To show this further, we included in Appendix G of the revised manuscript a complexity analysis of our method together with some GPU statistics during training (time vs accuracy, time vs allocated memory and time vs power consumption) on the GLUE benchmark. We hope the added results clarify your concerns about memory usage.
>
> 2. Thanks for pointing this out. A key advantage of FW being a projection-free method is its computational efficiency (see for example [Combettes & Pokutta, 2021]).
> In order to provide a clear theoretical analysis of the costs as compared to e.g. LoRA and AdaLoRA, we included in Appendix G a detailed discussion on this matter together with plots of real GPU usage.
> We appreciate the reviewer's interest in having an experimental evaluation on larger models such as Llama-2-7B. We are trying to set up the experiments and we will do our best to post here some (at least initial) results before the end of the discussion period, conditionally on our availability of computing resources.
>
> 3. Theorem 4.1 provides a bound on how much the error in the hypergradient is affected by the curvature of the problem. In practice, if $K$ is small in $S$, that is, the gradient is approximately locally constant,  then the error we perform with the closed-form solution $G$ for the hypergradient is small and decays at least linearly in $K$. All the assumptions in the statement of the theorem are necessary to understand how the curvature affects this error. However, we agree it is not obvious how to verify these assumptions in a practical setting. Nonetheless, empirically, we observed in our experiments that the approximate hypergradient estimation is a good proxy and behaves effectively.
>
> 4. $\tau$ was selected of order of the starting rank $O(r)$, leaving in this way potentially $O(1)$ for each entry of $s$. We added the precise value of $\tau$ used in the description of the experimental results in Sec7.
> What we do in practice is: motivated by the theoretical result in 6.5 about the identifiability of the optimal rank in a finite number of steps, we run Algorithm 1 for a small number of warm-up steps $n_0>0$ with the initial rank and then truncate $s_n$ (and $U$ and $V$ accordingly) when $n\geq n_0$ by removing the entries of $s_n$ that are smaller than the precision $\varepsilon$. In this way, we reduce to a problem with a smaller rank (as $s$ is now smaller and so are $U$ and $V$). The dimension of $s_n$ we have when the algorithm stops is the final rank $r^*$ of the model.
> In this way, the memory requirement is the same of AdaLoRa for a finite number of steps, and for the rest of the optimization the rank is reduced to $r^*$ (which as we can observe in Tabs 1,2, it often leads to much lower number of parameters).
> We apologize if this was not clear, we added a line for clarifying this point in the section containing the details of the algorithm.
>
> 5.  We are unsure here; the vector is already restricted to nonnegative values. Can you please clarify?
>
> [Combettes & Pokutta, 2021] Cyrille W Combettes and Sebastian Pokutta. Complexity of linear minimization
> and projection on some sets. Operations Research Letters, 49(4):565–571,
> 2021.

---

> > ### Comment · Reviewer_AG3z · 2024-11-26
> >
> > 1. By memory consumption, the reviewer refers to the peak memory usage observed during training, which often constitutes a critical bottleneck in the training of large language models (LLMs).
> >
> > 2. Could you elaborate on the time complexity and practical time consumption associated with the Frank-Wolfe direction search?

---

> > > ### Author Response · Authors · 2024-11-27
> > >
> > > 1. We thank the reviewer for the clarification. Since the comment was about the peak memory usage observed during training, we would like to remind you that we added plots in Appendix G of the revised manuscript concerning real GPU memory consumption and other statistics. As the plot shows, our effective memory usage on GPU is lower than AdaLoRA: this is due to its rank truncation procedure, which necessitates temporarily storing in memory the sensitivity measure, which effectively from a memory point of view is exactly a copy of the weight adapters.
> > >
> > > 2.  The feasible set of our problem is a simple polytope $S =  \textbf{conv}(0,e_1,\dots,e_r)$ with $e_i$ the $i$-th element of the canonical basis in $\mathbb{R}^r$.
> > > So the problem we need to solve in order to get our Frank-Wolfe direction (see Step 4 of Algorithm 2)   is the following linear program:
> > > $$ \quad z_n = \arg\min_{s \in S}\widetilde G(s_n)^\top s $$
> > > Taking into account the fundamental theorem of linear programming, the solution will thus be on a vertex of our polytope. By focusing on the vertices, we have $z_n=e_{i_n}$ with $i_n=\arg\min_i \nabla_i \tilde G(s_n) $, if there exists at least one component $i$ s.t. $\nabla_i \tilde G(s_n)<0$, and $z_n=0$ otherwise.
> > > \
> > > Therefore, the time complexity corresponds to finding the smallest entry in the hypergradient vector (that can be done simply by calling a torch.argmin function on $\widetilde G(s_n)$), which has a ${\cal O}(r)$ cost as highlighted in Appendix G. Notice that, a cost ${\cal{O}}(r)$ is negligible since the computation of the hypergradient itself has a complexity of ${\cal{O}}(Lrn^2)$, and in typical applications we have $n\gg r$ (the size of the original matrices is usually much bigger than the rank correction).
> > >
> > > We hope we were able to clarify your doubts, and if not please let us know.

---

### Official Review · Reviewer_G2LN · 2024-11-03

**Soundness:** 2
**Presentation:** 2
**Contribution:** 3
**Rating:** 6
**Confidence:** 2

**Summary:**

This paper introduces dEBORA, a bilevel optimization-based method that dynamically selects the optimal rank for each layer in parameter-efficient fine-tuning of large neural networks. By employing a hypergradient approximation that avoids large-scale computation, dEBORA significantly reduces the computational burden. Additionally, it integrates a stochastic away-step variant of the Frank-Wolfe algorithm, which eliminates the need for projection and ensures the identifiability of the optimal rank structure. Theoretical analysis confirms the convergence and optimality of the approach, while experimental results demonstrate that dEBORA outperforms existing low-rank adaptation methods in both efficiency and performance across various benchmarks.

**Strengths:**

- The paper presents a novel approach to low-rank adaptation by combining bilevel optimization with a dynamic rank-selection strategy, effectively addressing the challenge of parameter-efficient fine-tuning in large neural networks. Furthermore, the avoidance of implicit differentiation through an effective hypergradient approximation is a significant strength, as it reduces both computational costs and complexity.
- The experimental results are robust and cover a wide range of benchmarks.

**Weaknesses:**

- Despite providing several theoretical guarantees, the paper lacks clarity in the use of some symbols and definitions of variables, making it difficult for readers to follow the authors' proofs. For instance, the notation $\otimes$ first appears in line 141, while $\odot$ is introduced for the first time in line 215 without prior definition. Similarly, $\Delta$ is introduced in Equation (8) but is only defined in Theorem 6.2, and the variable $V$ is not defined in Equation (21). Additionally, the definition of $\mu$ is missing in Theorem 6.5. The authors are encouraged to carefully review the paper and provide clear definitions or explanations for variables and symbols, either upon their first appearance or in a comprehensive notation table.
- Although the overall structure of the paper is reasonable, the authors should consider removing the numbering from equations that are not referenced, as excess numbering creates a cluttered appearance. Furthermore, Equations (30) and (31) are noticeably smaller than the other equations, leading to inconsistencies in formatting. Additionally, the various variables within these equations lack definitions, making it challenging and time-consuming for readers. A careful review of whether each equation requires numbering, along with a focus on maintaining uniform formatting, would enhance the readability of the paper.
- While the authors mention the recent work BiLoRA (Qiang et al., 2024), no corresponding comparison is presented in the experiments. It would be valuable to understand the rationale behind this omission. Furthermore, while the authors state that they compare against the Pfeiffer adapter (Pfeiffer et al., 2021) and Houlsby adapter (Houlsby et al., 2019), the results do not clearly reflect these comparisons. Clarifying this point and providing results for these benchmarks would strengthen the experimental section.

**References:**

1. Rushi Qiang, Ruiyi Zhang, and Pengtao Xie. BiLoRA: A Bi-level Optimization Framework for Overfitting-Resilient Low-Rank Adaptation of Large Pre-trained Models. arXiv preprint arXiv:2403.13037, 2024.
2. Jonas Pfeiffer, Aishwarya Kamath, Andreas Rücklé, Kyunghyun Cho, and Iryna Gurevych. Adapterfusion: Non-destructive task composition for transfer learning, 2021. URL https://arxiv.org/abs/2005.00247.
3. Neil Houlsby, Andrei Giurgiu, Stanislaw Jastrzebski, Bruna Morrone, Quentin De Laroussilhe, Andrea Gesmundo, Mona Attariyan, and Sylvain Gelly. Parameter-efficient transfer learning for NLP. In Kamalika Chaudhuri and Ruslan Salakhutdinov (eds.), Proceedings of the 36th International Conference on Machine Learning, volume 97 of Proceedings of Machine Learning Research, pp. 2790–2799. PMLR, 09–15 Jun 2019. URL https://proceedings.mlr.press/v97/houlsby19a.html.

**Questions:**

1. The paper introduces a strategy for dynamically selecting the optimal rank $r$ for each layer. However, it is unclear how rank $r$ is adjusted after determining the optimal solution $s$. Is $r$ recalibrated based on the optimal $s$, or is there a specific strategy for automatically tuning $r$? Could the authors clarify this process in detail?

2. In Theorem 4.1, the authors state, "assume that the gradient is locally approximately constant." However, the subsequent equations appear to be formulated in terms of the Hessian. Could the authors clarify the relationship between these two concepts?

3. Regarding equation (3), if the matrix $\mathcal{B}$ is constrained to lie on a manifold—as mentioned later in the experiments with the Oblique and Stiefel manifolds—would the first-order optimality condition still hold as stated? This consideration could significantly impact the approximations and key results presented in the paper. Could the authors clarify how the manifold constraints influence the formulation of the first-order optimality condition?

4. In equation (35), is it necessary for the authors to clarify the invertibility of the product $AB$? Providing a discussion on this aspect would enhance the understanding of the conditions under which the equation holds.

5. In line 755, the notation $||\partial_{\mathcal{B}}f_{2}^{\*}||$ should be revised to $||\partial_{\mathcal{B}}f_{1}^{\*}||$. Additionally, could the authors clarify where the boundedness of this term originates?



The article contains several typographical and punctuation errors that require careful review by the authors. For example:

- Punctuation errors: in lines 208, 240, 319, 401, and 407, there are missing commas. In line 686, there is an extra period.
- Line 679: "By" should be "by".

---

> ### Author Response · Authors · 2024-11-15
>
> We wish first of all to thank the reviewer for all the comments. Below, we provide a point-by-point response to the weaknesses and questions raised.
>
> * Weaknesses:
> 1. We apologize for the confusion with the notation.   $\otimes$ refers to the outer product of tensors, and $\odot$ to the Hadamard product (entrywise);  we assumed this was a standard notation, but we agree it would be much better to clarify. We now define both of these operations at their first occurrence.
> About $\Delta$, we now define it at the beginning of Section 6, right after the description of the optimization problem (4). For what concerns $V$, we added a sentence with the definition before its first appearance.
> The $u$ in theorem 6.5 was just a typo; we deleted it in the updated version.
> 2. This is a fair point which we agree with.  We removed the numbering from each equation that is not referenced in the text.
> 3. Thanks for raising this point. Concerning Pfeiffer and Houlsby adapters, we apologize for this. We simply accidentally forgot to add them to the table, and we realized that only after the submission deadline. We added them in the revised version.
> We cited BiLoRA for fairness as it is directly related to our approach. However, their implementation is not public yet, and we were not able to reproduce their results. Also, please note that they calculate the hypergradients using implicit differentiation; thus, their method is computationally strictly more demanding than ours.
>
> * Questions:
>
> 1. We apologize that this was not clear. What we do in practice is: we run Algorithm 1 for a small number of warm-up steps $n_0>0$ with the initial rank and then truncate $s_n$ (and $U$ and $V$ accordingly) when $n\geq n_0$ by removing the entries of $s_n$ that are smaller than the precision $\varepsilon$. In this way, we reduce to a problem with a smaller rank (as $s$ is now smaller and so are $U$ and $V$). The dimension of $s_n$ we have when the algorithm stops is the final rank $r^*$ of the model.
> As discussed with reviewer 1wqr, this procedure is theoretically justified by Theorem 6.5, which guarantees the identifiability of the optimal rank (face of the $L^1$ simplex) in a finite number of steps.
> We have modified Algorithm 1 by adding step 10, which clarifies this point, and have added a sentence to clarify this in the description on line 306.
>
> 2. The subsequent equation is there to clarify what "locally approximately constant gradient" means. In other terms, we require the Hessian to be small enough in the ball $S$. These two concepts are related as a zero Hessian on a connected set would imply that the gradient is a constant function (and vice-versa). In Theorem 4.1 we just make explicit the control on the error based on the norm of the Hessian.
>
> 3. This is a fair point. For the sake of simplicity, we decided to limit the theoretical discussion to the Euclidean case. However, everything transfers with minor adjustments to the Riemannian setting by interpreting all the derivatives through their Riemannian counterpart.
> A similar derivation is done for example in [Li & Ma, 2024] and the computations in our case would be similar to those presented there.
>  In particular, in our setting the only Riemannian manifold is in the lower-level problem. In this case the stationarity condition should be interpreted as $\partial_{\mathcal B} f_2(\mathcal B^*(s),s) = 0$ where $\partial_{\mathcal B}f_2(\mathcal{B}^*(s),s)$ is now the differential restricted to the tangent space $T_{\mathcal B^*(s)} \mathcal V$ to the manifold $\mathcal V$ at the point $\mathcal B^*(s)$, with $\mathcal V$ either the Stiefel or the oblique manifold in our case. The implicit gradient equation (3) is derived in the same way by interpreting the Hessian as the Riemannian Hessian on $\mathcal V$. We thank the reviewer for pointing out this potential source of confusion. We now clearly state this at the end of Section 3.
>
> 4. Apologies, but we could not find the inverse of $AB$ in the proof; we assume the reviewer is referring to $CD$, whose invertibility is actually one of the assumptions of the theorem.
>
> 5. Thanks to the reviewer for noticing this typo. Concerning the boundeness, Oblique of Stiefel manifolds are compact, thus their boundeness follows immediately from continuity.
>
> 6. We thank the reviewer again, we corrected the typos in the revised version.
>
>
> [Li & Ma, 2024] Jiaxiang Li and Shiqian Ma. Riemannian bilevel optimization. arXiv preprint
> arXiv:2402.02019, 2024

---

> > ### Comment · Reviewer_G2LN · 2024-11-17
> >
> > Thank you for the timely and patient responses. All of my concerns have been thoroughly addressed, and I greatly appreciate the efforts made. I will revise my scores.

---

### Meta-Review · Area_Chair_a7Gb · 2024-12-22

**Metareview:**

This paper introduces dEBORA, a bilevel optimization-based method that dynamically selects the optimal rank for each layer in parameter-efficient fine-tuning of large neural networks. By employing a hypergradient approximation that avoids large-scale computation, dEBORA significantly reduces the computational burden. Additionally, it integrates a stochastic away-step variant of the Frank-Wolfe algorithm, which eliminates the need for projection and ensures the identifiability of the optimal rank structure. Theoretical analysis confirms the convergence and optimality of the approach, while experimental results demonstrate that dEBORA outperforms existing low-rank adaptation methods in both efficiency and performance across various benchmarks.

Selected strengths:
- The paper presents a novel approach to low-rank adaptation by combining bilevel optimization with a dynamic rank-selection strategy, effectively addressing the challenge of parameter-efficient fine-tuning in large neural networks. As such, the paper addresses a longstanding issue in LoRA: selecting the optimal rank for each low-rank adapter, a key factor in balancing model performance and computational efficiency.
- The experimental results are robust and cover a wide range of benchmarks. With fewer parameters, the proposed method demonstrates competitive or superior performance compared to existing approaches such as LoRA and AdaLoRA.
- An interesting new application of the Frank-Wolf method.

Selected weaknesses:
- Lack of clarity in the use of some symbols and definitions of variables, which makes it difficult for readers to follow the proofs. Some formatting/style recommendations were made by the reviewers.
- While the authors mention the recent work BiLoRA, comparison to this method was not done in the experiments.
- While selecting the optimal rank is indeed critical for LoRA, in many practical applications, concerns about memory consumption and training time are often more pressing. The proposed optimization approach suggests a reduction in parameter count, but the practical impact on memory usage remains unclear.
- Is the method applicable to even larger models?
- Some questions regarding Theorem 4.1
- Several more minor issues were raised

All three reviewers recommended acceptance, with scores 6, 6, and 8. I have read the reviews, rebuttals, and discussion. I believe the responses were reasonable. The paper is in a good shape to be accepted to the conference.

**Additional Comments On Reviewer Discussion:**

Several questions were asked by the reviewers (e.g,  it is unclear how rank is adjusted after determining the optimal solution $s$; questions about notation, theory, assumptions). The key questions were answered satisfactorily.

---

### Decision · Program_Chairs · 2025-01-22

Accept (Poster)